# Age Associated Decrease of MT-1 Melatonin Receptor in Human Dermal Skin Fibroblasts Impairs Protection Against UV-Induced DNA Damage

**DOI:** 10.3390/ijms21010326

**Published:** 2020-01-03

**Authors:** Kelly Dong, Earl Goyarts, Antonella Rella, Edward Pelle, Yung Hou Wong, Nadine Pernodet

**Affiliations:** 1Skin Biology & BioActives, R&D, Estée Lauder Companies, Melville, NY 11747, USA; egoyarts@estee.com (E.G.); arella@estee.com (A.R.); pelle623@yahoo.com (E.P.); npernode@estee.com (N.P.); 2Division of Life Science and the State Key Laboratory of Molecular Neuroscience, Hong Kong University of Science and Technology, Hong Kong, China; boyung@ust.hk

**Keywords:** aging, circadian rhythm, melatonin, skin fibroblasts, ultraviolet (UV) irradiation

## Abstract

The human body follows a physiological rhythm in response to the day/night cycle which is synchronized with the circadian rhythm through internal clocks. Most cells in the human body, including skin cells, express autonomous clocks and the genes responsible for running those clocks. Melatonin, a ubiquitous small molecular weight hormone, is critical in regulating the sleep cycle and other functions in the body. Melatonin is present in the skin and, in this study, we showed that it has the ability to dose-dependently stimulate *PER1* clock gene expression in normal human dermal fibroblasts and normal human epidermal keratinocytes. Then we further evaluated the role of MT-1 melatonin receptor in mediating melatonin actions on human skin using fibroblasts derived from young and old subjects. Using immunocytochemistry, Western blotting and RT-PCR, we confirmed the expression of MT-1 receptor in human skin fibroblasts and demonstrated a dramatic age-dependent decrease in its level in mature fibroblasts. We used siRNA technology to transiently knockdown MT-1 receptor in fibroblasts. In these MT-1 knockdown cells, UV-dependent oxidative stress (H_2_O_2_ production) was enhanced and DNA damage was also increased, suggesting a critical role of MT-1 receptor in protecting skin cells from UV-induced DNA damage. These studies demonstrate that the melatonin pathway plays a pivotal role in skin aging and damage. Moreover, its correlation with skin circadian rhythm may offer new approaches for decelerating skin aging by modulating the expression of melatonin receptors in human skin.

## 1. Introduction

Skin, the largest and the most external organ of the human body, is directly exposed to environmental insults including extreme temperatures, light, humidity, ultraviolet (UV) radiation, air pollution (smog, ozone, particulate pollutants, etc.) and pathogens. Like other tissues and organs in our body, the skin also follows a circadian rhythm. This 24 h cycle, where cellular daytime functions differ radically from nighttime functions, is required to maintain a healthy skin by managing the energy supply and cellular conditions for optimal protection and repair. Circadian rhythm helps to synchronize the various skin cell types with each other as well as with the natural rhythm of the human body.

Many of the environmental factors which cause oxidative stress on the skin can result in cell damage. The oxidative stress, especially by UV radiation, generates reactive oxygen species (ROS) that cause damage to cellular proteins, lipids, nucleic acids and membranes, as well as to organelles such as mitochondria [1]. Although the skin is equipped with an elaborate antioxidant system to deal with oxidative stress, chronic exposure to environmental insults, especially UV exposure, can overcome the endogenous antioxidant capacity of the skin and result in cellular damage and skin disease [2,3]. Many of these antioxidants and antioxidant enzyme systems in the skin are reduced with age, making the older skin more vulnerable to environmental insults [4]. The skin produces several protective molecules including melatonin, melanin and vitamin D to counteract oxidative stress [4,5]. The production of these molecules might be controlled by a 24 h light/dark cycle. In addition, melatonin and vitamin D regulate some aspects of the circadian rhythm and the redox state of the skin [6]. Under normal environmental conditions, many attributes of the human skin follow a periodicity, they include hydration and trans-epidermal water loss (TEWL), capillary blood flow, sebum production, skin temperature, surface pH, keratinocyte proliferation rates and even the visibility of facial rhytides [7,8,9,10,11]. It is clear that all major cell types in the human skin have functional circadian machinery and display specific periods and phase relationships in gene expression, suggesting regulatory mechanisms that are particular to each cell type [12].

Melatonin, the main hormone released by the pineal gland, is critical for regulating the sleep cycle by modulating the circadian clock. Melatonin is also synthesized in many tissues other than the pineal gland, including human skin [13,14]. Here the process starts with tryptophan that is converted to serotonin via tryptophan hydroxylase and aromatic amino acid decarboxylase activity. Arylalkylamine *N*-acetyltransferase (AANAT) mediates the conversion of serotonin to *N*-acetylserotonin which ultimately is transformed to melatonin via hydroxyindole-*O*-methyltransferase. AANAT is considered to be the rate limiting step in melatonin production and follows a rhythm [14,15].

Skin cells not only produce melatonin but also express melatonin MT-1 and MT-2 receptors [16]. Melatonin binds to MT-1 and MT-2 receptors, G-protein coupled membrane bound receptors, which are present in the suprachiasmatic nucleus (SCN), creating biological rhythms that induce sleep [17]. 

MT-1 and MT-2 receptors are important for protection against environmental insults to the skin as well as for the production of vitamin D [18]. They are also involved in the regulation of many physiological systems such as endocrine, reproductive, cardiovascular and immune-system, as well as in skin pigmentation, hair growth, carcinogenesis and aging [14]. MT-1 and MT-2 were also previously reported to be essential for melatonin’s DNA repair ability via p53 [19].

Melatonin is also able to suppress UV-induced damage to skin cells and shows strong antioxidant activity in UV-exposed cells, counteracting the generation of ROS and protecting cells from mitochondrial and DNA damage [20]. We have previously reported that melatonin increases *PER1* levels in normal human keratinocytes, which means that it is directly involved in controlling the circadian rhythm of skin cells [16]. With regard to the 24 h light/dark cycle, melatonin is highest in the evening where it influences *PER1* gene expression in skin. Taken together, there is considerable support for melatonin to be a beneficial compound for human skin [2,6,7,13,21,22,23,24].

Aging and the associated decline in circadian rhythm can elevate oxidative stress through the increased production and accumulation of ROS [11,25]. Melatonin levels decline with age, further contributing to a decline in the antioxidant capacity of the skin. The decrease in melatonin is associated with the intrinsic dysregulation of circadian rhythm with age. Environmental exposure of the skin to extrinsic factors such as solar radiation also elevates the level of oxidative stress. Therefore, in this study we evaluated the effect of age on the ability of melatonin to protect human skin fibroblasts from UV-induced cellular damage. We found that there was an age-dependent decrease of MT-1 receptor in aged human fibroblasts and that suppressing melatonin receptor temporarily in vitro increased H_2_O_2_ production and potentiated the UV-induced DNA damage in human skin fibroblasts. We propose that this age-dependent reduction in melatonin receptor, concomitant with a reduction in melatonin synthesis, result in a higher propensity for cellular damage and a loss of repair in the skin. This presents an opportunity for the stimulation of MT-1 receptor as a useful strategy for improving overall skin health.

## 2. Results

### 2.1. Melatonin Stimulates PER1 Clock Gene in Normal Human Dermal Fibroblast (NHDF) and in Normal Human Epidermal Keratinocytes (NHEK)

Clock gene activity in the skin is modulated by several factors. Melatonin is a critical molecule, which is increased at nighttime and distributed throughout the whole body. It is also present in skin where it has been shown to support skin protection. During a normal circadian cycle, melatonin is highest in the evening [26]. Melatonin in turn, stimulates the *PER1* circadian clock gene expression in human skin cells [16]. In this study we evaluated the dose response of melatonin for increasing *PER1* expression in human dermal fibroblast and in human epidermal keratinocytes. NHDF and NHEK transfected with a *PER1* luciferase reporter construct, were treated with different concentration of melatonin after transfection and the level of luciferase activity measured. The generation of bioluminescence was used as a surrogate marker for *PER1* transcription. As can be seen from Figure 1, there is an increase of RLU (relative lumens) or *PER1* expression in response to melatonin in NHDF and NHEK. At a dose of 200 µM of melatonin, a 2 to 3-fold stimulation of *PER1* expression was observed in NHDF and NHEK. 

### 2.2. NHDF Express MT-1 Receptor and Its Level Is Decreased with Age

In order to gain further insight into the melatonin activation pathway, we evaluated the level of MT-1 receptor in normal human NHDF. Melatonin interacts with two G protein-coupled plasma membrane receptors, MT-1 and MT-2, through which melatonin mediates its cellular effects. Since endogenous melatonin levels decrease in human skin as a function of age, it was of interest to evaluate whether this decrease is also accompanied by a reduction in MT-1 melatonin receptor.

Levels of melatonin MT-1 receptor were evaluated by immunocytochemistry in NHDF derived from young (19-year-old) and aged (67-year-old) NHDF. As can be seen from Figure 2A, MT-1 can be visualized in young NHDF. However, the level of MT-1 receptor was dramatically reduced in aged NHDF. In the immunofluorescence images, MT-1 was barely visible in the older cells. This result clearly suggests that the level of MT-1 receptor in skin decrease with age.

Western blotting was used to correlate the immunostaining of the MT-1 monomer with the surface staining of the MT-1 receptor on NHDF. In Figure 2B, a band is present around 60 kDa that correspond to the MT-1 receptor [27]. Finally, a 4-fold decrease in MT-1 mRNA expression was observed between young and old NHDF in Figure 2C.

### 2.3. Characterization of MT-1 Receptor Knockdown Using siRNA Technology

Using siRNA technology against MT-1, we evaluated the role of this melatonin receptor in young 19-year-old NHDF. A reduction of MT-1 mRNA levels was achieved through electroporation with specific siRNA. Western blotting was used to correlate this decrease in MT-1 transcripts with a decrease in MT-1 monomers. As seen from Figure 3A the protein band present in the MT-1 KD and, corresponding to MT-1 receptor, is about 42% less intense than the band of the non-targeting siRNA control (NT). 

This experiment demonstrated that the expression of MT-1 receptor can be transiently inhibited in human NHDF. When normalized relative to the housekeeping gene, GAPDH, the efficiency of knockdown for the MT-1 mRNA expression was about 50%, indicating a significant reduction of functional MT-1 transcripts in these cells (Figure 3B). This inhibition can extend beyond 48 h, enabling us to study the protective role of melatonin receptors against environmental stress such as UV irradiation. 

### 2.4. Melatonin Receptor Knockdown Does not affect the Cell Viability of NHDF

Since cells are subjected to stressful conditions during electroporation, it may lead to impaired cell viability and hence sensitization to UV-induced DNA damage in the siRNA treated cells. To exclude this possibility, cell viability of NHDF was evaluated after the cells were exposed to non-targeted or targeted siRNA to MT-1 receptor for 72 h. As shown in Figure 3C, cell viability was not affected by the treatment of NHDF with 300 nM siRNA. This confirms that the specific UV response of siRNA targeted NHDF was indeed due to the knockdown effect of MT-1 receptor and not due to any non-specific cell viability effect.

### 2.5. MT-1 Receptor Knockdown Increases the Sensitivity of NHDF to UV Irradiation

When NHDF derived from a young subject (which contain more MT-1 receptor) were exposed to UV (5 J/cm^2^ UVA + 40 mJ/cm^2^ UVB), DNA damage was clearly demonstrated by the comet assay 4 h after UV irradiation. In NHDF whose melatonin receptor was transiently knocked down, the DNA damage was dramatically increased. As demonstrated in the graph shown in Figure 4A and in the fluorescent microscopy image in Figure 4B, non-irradiated NHDF showed only minimal DNA damage, with or without siRNA (both non-targeted and targeted siRNA) exposure. However, NHDF exposed to UV demonstrated dramatic increases in DNA damage (420% over non-exposed cells). UV-induced DNA damage in NHDF treated with non-target siRNA was slightly increased (~20%) but the change was insignificant. In contrast, DNA damage in MT-1 knockdown NHDF showed a very significant increase versus controls, showing that they are more prone to UV-induced damage. These results clearly demonstrate an important role of MT-1 receptor in helping to reduce DNA damage.

### 2.6. Increased Oxidative Stress in MT-1 Receptor Knockdown NHDF

UV exposure of human skin NHDF results in higher H_2_O_2_ levels. In the study, MT-1 receptor in NHDF was knocked down using the siRNA technology. 72 h after the knockdown, cells were exposed to UV (UVA + UVB). 6 h after UV exposure, the amount of H_2_O_2_ generated in the cells was measured and normalized to the total protein content. As seen from Figure 5, the amount of H_2_O_2_ generated in the control cells and cells exposed to non-targeted siRNA, showed insignificant stimulation of H_2_O_2_ production in response to UV exposure (15% and 0% respectively). However, MT-1 knockdown NHDF that were exposed to UV demonstrated a substantial increase in H_2_O_2_ production compared to both non-targeted siRNA cells and untreated cells. The knockdown cells generated 54% more H_2_O_2_ production than the untreated cells and 74% more than the non-targeted siRNA cells, suggesting a significant UV-induced damage response and ROS generation in the MT-1 knockdown cells.

## 3. Discussion

Human body responds to a night/day rhythm through entrainment of the clock genes. At a cellular level the circadian clock mechanism is regulated by interdependent feedback loops of transcription and translation of specific protein repressors. Most cells in the body express autonomous clock gene family members which give rise to a homeostatic circadian rhythm when young and healthy [5,13,28].

There is accumulating evidence that dysregulation of the clock mechanisms leads to increased skin damage, which are associated with the generation and accumulation of ROS [29]. In fact, near lethal oxidative stress has been reported to cause resetting of the clock and synchronization in cultured murine fibroblasts [30]. Inflammatory markers are elevated when skin cells are desynchronized. Melatonin has been shown to play a direct role in controlling *PER1* clock gene expression and the natural secretion of melatonin follows a circadian rhythm, highest at night and lower during the day. Melatonin secretion is sensitive to light, therefore changes in the light/dark cycle leads to significant changes in melatonin release. When circadian rhythm is dysregulated, the endogenous levels of melatonin will be perturbed and exacerbate the effects of UV to skin [11]. Aging itself is known to result in increased oxidative stress in most organs and tissues but the exact mode of action of circadian clocks in this aging mechanism is not fully understood [31].

The connection between UVB and the generation of ROS is well established [32,33,34,35,36]. In our previous studies, exposure of synchronized cells to UVB disrupted ATP synthesis and caused a temporal release of hydrogen peroxide [31]. These data support the perceived connection between clock gene activity, oxidative stress and skin aging [32,37]. Skin aging is correlated with a reduction of protective mechanisms against oxidative stress, a desynchronization of cells from their natural circadian rhythm and a disruption of endogenous melatonin levels [38].

In addition to playing an important role in regulating circadian rhythm, melatonin has many other biological effects, including the upregulation of the antioxidant enzymes catalase, glutathione peroxidase and superoxide dismutase [39]. In keratinocytes, melatonin attenuates the amount of DNA damage caused by UV radiation [40,41]. Since melatonin is also produced in skin, it has been shown to be involved in oxidative stress control and we have shown its relation with *PER1* clock gene level, it may have clinical and cosmetic benefits in preventing and treating skin conditions. Recent studies have provided considerable evidence that suggests melatonin or melatonin-like molecules may be immensely useful compounds to impede skin aging, especially due to their ability to counteract the oxidative stress [13]. The present demonstration of melatonin and its receptors in regulating the circadian rhythms in skin cells further strengthens this notion.

In this study we have demonstrated that melatonin, whose levels in skin are controlled by circadian rhythm, itself regulates the expression of clock gene, *PER1*, suggesting a feedback loop mechanism operating in human skin cells. The effects of circadian rhythm on skin physiology are already well documented. Skin functions, such as, increased skin thickness, better barrier function, higher sebum production, high pH and low cell proliferation are reported to be highest in day time, while DNA repair, cell proliferation, skin temperature, barrier permeability, skin penetration and skin blood flow are highest at night time [11].

Since most of the actions of melatonin are mediated via its two receptors, MT-1 and MT-2 and that melatonin level itself is decreased in skin with age, we evaluated the relationship of MT-1 melatonin receptor to skin aging using NHDF derived from young and old subjects. We also evaluated the role played by MT-1 melatonin receptor in mediating the well-known age-dependent increased sensitivity of skin cells to UV irradiation and UV-induced damage. We observed significant decrease in MT-1 receptor in NHDF derived from aged donors. Importantly, MT-1 knockdown NHDF demonstrated enhanced sensitivity to DNA damage in response to UV irradiation. In addition, MT-1 knocked down NHDF also showed significant stimulation of ROS and H_2_O_2_ productions in response to UV irradiation.

Recently it was reported that the photoprotective effect of melatonin was independent of the MT-1 receptor in human epidermal melanocytes [24]. A plasma membrane transporter could deliver the melatonin to the cytoplasm [42]. Once inside the cell, melatonin and its metabolites can induce NRF2 transcription in the nucleus, possibly inhibit the proteasome [43] or become sequestered by the mitochondria [44,45]. We do not know which of these mechanisms, as well as others not considered, are sufficient for the photoprotective effect in cell culture. In our fibroblast model, MT-1 receptor knockdown could make the mitochondria susceptible to increase oxidative stress if an automitocrine system were present in the fibroblast. Further experiments are needed to explain why a reduction in the MT-1 receptor results in more DNA damage from UV while a high dose of melatonin prevents DNA damage.

Negative impact of UVB on clock gene expression in normal human keratinocytes has been previously demonstrated, leading to low energy production and high oxidative stress [46]. Our results are consistent with these previous observations. In addition, our results point to the importance of melatonin, along with MT-1 receptor, in the control of skin circadian rhythm which may have a direct impact on the ability of skin cells to control damage.

In a recent study we have established an association between chronic poor sleep quality and the function, integrity and appearance of skin in human subjects [46]. Chronic poor-quality sleep, in the “desynchronized” group, is associated with accelerated intrinsic skin aging and less effective skin barrier function. Good quality sleep, in the synchronized group, was associated with more efficient recovery from UV-induced erythema and better self-perception of overall appearance. These observations highlighted the importance of synchronization with circadian rhythm for optimal skin functions at the cellular level. The mechanism through which melatonin and MT-1 receptor modulate and minimize environmental damage to skin such as UV-induced damage maybe directly mediated through the control of circadian rhythm, which would help to slow down the aging process in cells. In summary, our data demonstrate a pivotal role played by melatonin and MT-1 receptor in skin aging and suggests new targets for improving skin aging by modulating melatonin receptors expression in human skin.

## 4. Materials and Methods

### 4.1. Cell Culture

Normal human epidermal keratinocytes (NHEK) stocks were sub-cultivated at the second passage in full EpiLife media. Normal human dermal fibroblasts (NHDF) derived from a 62-year-old donor, a 67-year-old donor and a 19-year-old donor were obtained from the Coriell Institute (Camden, NJ, USA) and grown on 4-well chamber slides for immunocytochemistry or on 60 mm dishes for electroporation and UV exposure. Cells were sub-cultivated in Dulbecco’s Modified Eagle’s Medium (DMEM) with 10% bovine calf serum and 1% penicillin/streptomycin at 37 °C in a 5% CO_2_ humidified incubator and used at approximately passage 8.

### 4.2. PER1 Gene Expression

NHEK (5 × 10^4^ cells) were plated onto a black walled, 96 well microtiter plate for 3 h in EpiLife media. The cells were then transfected in supplement free media with a plasmid that contained luciferase as the reporter gene ligated downstream to a *PER1* promoter element (*PER1* promoter gene 71575, cat# C579177). This plasmid construct was synthesized for Estee Lauder by DNA 2.0 (Palo Alto, CA, USA). Cells transfection was facilitated by the addition of Plus Reagent and Lipofectamine reagent (Thermo Fisher Scientific, Waltham, MA, USA). After additional 4 h in supplemented free media, full media or full media with different concentration of melatonin (100–200 µM) was added to the cells. Lastly, a luciferin-containing reagent, Bright-Glo (Promega, Madison, WI, USA), was added and luminescence measured at 470 nm in a LMax luminometer (Molecular Devices, San Jose, CA, USA).

NHDF (10^6^ cells) were electroporated with the above plasmid. The electroporation of the cells was facilitated by the addition of the Nucleofector Solution and Supplement following the manufacturer’s protocol (Amaxa Human Dermal Fibroblast Nucleofector Kit, Lonza, Basel, Switzerland, Cat No. CC-2509). After electroporation, 4 × 10^4^ cells per well were plated onto a black walled, 96-well microtiter plate for overnight in DMEM media. The next day the cells were washed with HBSS and full media or full media with different concentration of melatonin (100–200 µM) was added and the cells were incubated for approximately 24 h. As described previously, Bright-Glo was used to measure luminescence in a LMax luminometer (Molecular Devices).

### 4.3. MT-1 Melatonin Receptor Immunohistochemistry 

NHDF from young and old donors in culture were rinsed with Dulbecco’s phosphate buffered saline (DPBS), fixed in 4% paraformaldehyde prepared in DPBS and perforated with 0.05% Triton in DPBS for 5 min. Cells were blocked for 40 min with 1% bovine serum albumin in DPBS. Cells were probed for the expression of MT-1 melatonin receptor with specific antibodies directed against MT-1 (sc13179) (Santa Cruz Biotechnology, Dallas, TX, USA) overnight at 4 °C followed by a 1 h incubation with a fluorescent FITC secondary antibody (Santa Cruz Biotechnology). Cells were also stained for the nucleus with DAPI (4′,6-diamidino-2-phenylindole) and preserved by the addition of Vectashield mounting medium (Vector Laboratories; Burlingame, CA, USA). Confocal microscopy images were captured with a Nikon A1 confocal microscope and a 20× objective with a pinhole radius setting of 21.71 and a laser power of 140 for DAPI and 39 for FITC.

### 4.4. Western Blotting

NHDF cells were lysed with CytoBuster protein extraction reagent (EMD Millipore Corp., Billerica, MA, USA) and used according to the manufacturer’s protocol. The extraction was done in the presence of Halt Protease Inhibitor Cocktail (Thermo Scientific, Rockford, IL, USA) at RT for 5 min. Total proteins were recovered from adult NHDF, 19-years old and 62-years old and from 72 h post electroporation of 19-year-old NHDF with siRNA. Total cell lysate was recovered after centrifugation (16,000× *g*) at 4 °C. The supernatants were separated from the pellets and assayed using the Micro BCA protein assay kit (Thermo Scientific, Rockford, IL, USA). Protein concentration was measured on a SpectraMAX 190 spectrophotometer (Molecular Devices) and analyzed with SoftMax Pro, version 5.4.1. Protein electrophoresis and Western blotting was done with Novex gels and Novex electrophoretic chambers. Approximately 30 μg of protein were loaded on precast NuPAGE 4–12% Bis-Tris gels. Protein was transferred for 1.5 h at RT onto Immobilon-FL membranes (EMD Millipore, Bedford, MA, USA) in the presence of 10% methanol. A pre-stained protein ladder called Chameleon Duo (LI-COR, Lincoln, NE, USA) was used as a molecular weight marker. The primary antibody was goat polyclonal, ab87639 (Abcam, Cambridge, MA, USA). The secondary was an IRDye 680 Donkey anti-goat antibody (926-68074). Western detection methods were followed according to the manufacturer, LI-COR. Blocking conditions required Odyssey blocking buffer (927-40000, LI-COR) and 0.2% Tween (P7949, Sigma-Aldrich, St. Louis, MO, USA). Visualization of the bands was done with the Odyssey CLx instrument. 

### 4.5. Electroporation of siRNA 

Trypsinized young NHDF from a 19-year-old donor were cultured in DMEM to passage 8. The cells were removed from the medium, washed with Hanks buffered saline solution and electroporated with the AMAXA nucleofector II device, using 10^6^ NHDF in 100 µL of electroporation buffer containing siRNA. siRNA reagents were purchased from Thermo Fisher Scientific; they include siRNA directed against the melatonin MT-1 receptor (S9049) and the non-targeting siRNA control (4390843). NHDF received 300 nM of siRNA during electroporation. Electroporated cells were plated on 100 mm plates for RNA purification or on 6-well plates with glass inserts and collagen coated (MatTEK, Ashland, MA, USA) for immunostaining of MT-1 melatonin receptor. Two days post electroporation, RNA was recovered to determine the level of knockdown by RT-PCR. Three days post electroporation, the 19-year-old NHDF were recovered for Western blotting. 

### 4.6. Real-Time PCR

RNA was recovered via column purification using RNeasy kit (Qiagen, Hilden, Germany) and cDNA prepared with iScript (BioRad, Hercules, CA, USA). cDNA was amplified by PCR (50 cycles). Taqman primer/probes were used to detect the amplified PCR product in a Quant Studio 7 Flex Realtime PCR instrument (ThermoFisher Scientific). The primer/probes (Thermo Fisher Scientific) include Hs99999905_m1 for GAPDH (glyceraldehyde-3-phosphate dehydrogenase) and Hs00195567_m1 for the human MT-1 receptor.

### 4.7. UV Exposure

Cells were washed once with DPBS and then irradiated through a thin layer of DPBS in the Dr. Gröbel irradiation chamber fitted with UVA (F15T8BLB) and UVB (G15T8E) bulbs (Sanko Denki; Torrance, CA, USA). Spectra can be seen in Appendix A. The quantity of exposure was monitored by a UV monitor and the amount of UV exposure was 5 J/cm^2^ of UVA and 50 mJ/cm^2^ of UVB [47] for the H_2_O_2_ assay and 5 J/cm^2^ of UVA and 40 mJ/cm^2^ for the comet assay.

### 4.8. Comet Assay (DNA Damage)

The comet assay (single-cell gel electrophoresis) was used to measure DNA strand breaks in NHDF. NHDF in DPBS were exposed to a combination of UVA (5 J/cm^2^) and UVB (40 mJ/cm^2^) using the appropriate bulbs in the Dr. Gröbel irradiation chamber. Cells were lysed with detergent and high salt and then embedded in agarose on a microscope slide. Electrophoresis at high pH results in structures called nucleoids containing supercoiled loops of DNA linked to the nuclear matrix. These structures resemble comets. The comets were visualized under a microscope after staining with SYBR Gold (Thermo Fisher Scientific, #11494). Comets were evaluated with comet scoring software (TriTek CometScore™ v1.5) that determined the tail moments of each comet which correlated to DNA damage. The software determined the tail moment by taking the fluorescence intensity of the comet tail relative to the head and the overall length of the comet into consideration. Generally, a longer comet and a tail that is more intense than the head is indicative of a larger amount of DNA damage.

### 4.9. Cell Viability Measurement

Culture medium was removed from the cells in 96-well plates and replaced with 100 µL of 10% Alamar blue (Thermo Fisher Scientific, #DAL1100) solution in DMEM. The cells were incubated with the Alamar blue solution for 2 h and then the fluorescence was read on a fluorescent plate reader at 530 nm excitation and 590 nm emission. These values were normalized to protein levels.

### 4.10. H_2_O_2_ Measurement in UV-irradiated NHDF

To make a 500 µg/mL stock solution, 50 µg of 5-(and-6)-chloromethyl-2′,7′-dichlorodihydrofluorescein diacetate, acetyl ester (Thermo Fisher Scientific, #C6827) was dissolved in 100 µL of ethanol. A 5 µg/mL working solution was made by diluting 80 µL of the stock solution in 8 mL DPBS. Following exposure to UV, cells were washed once with DPBS and then 100 µL of the working solution was added to each well. Cells were returned to the incubator for 20 min. Then 100 µL of 25 mM NaN_3_ was added to each well and the plate was returned to the incubator for another 2 h. The fluorescence was read on a fluorescent plate reader at 490 nm excitation and 520 nm emission. These values were normalized to protein levels. Protein quantification was carried out using Micro BCA Protein Assay Kit (Thermo Fisher Scientific, # 23235). Reagents from the H_2_O_2_ assay were removed and 50 µL of lysis buffer (0.1N NaOH) were incubated in each well on the shaker for 5 min. The working solution from the Micro BCA Protein Assay Kit was prepared by combining 5 mL of Reagent A, 4.8 mL of Reagent B and 200 µL of Reagent C. 50 µL of the working reagent were added to the each well containing cells/lysis solution. The plate was placed into the incubator for 120 min and then read on a spectrophotometer at 562 nm.

### 4.11. Statistics 

One-way ANOVA and a Bonferroni post-test was used to analyze the data in GraphPad Prism 8. Error Bars in all graphs are SEM.

## Figures and Tables

**Figure 1 ijms-21-00326-f001:**
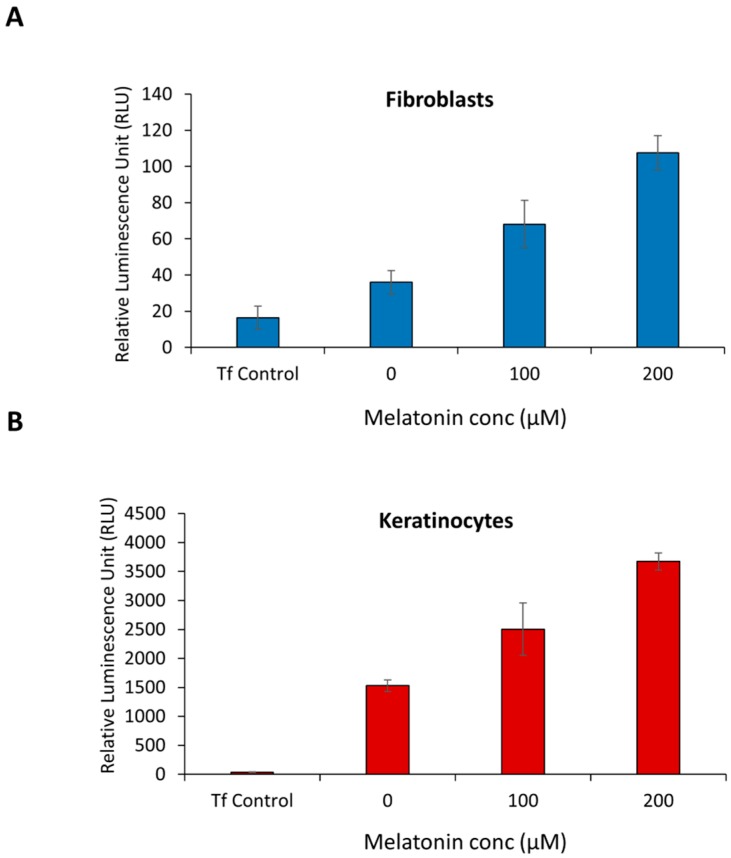
*PER1* expression increases in response to higher concentration of melatonin. (**A**) Normal Human Dermal Fibroblasts (NHDF) and (**B**) Normal Human Epidermal Keratinocytes (NHEK) were incubated with different concentration of melatonin for 24 h, and the level of expression of PER1 was evaluated using a reporter gene assay. Tf Control, transfection control. Error bars are SEM. *N* = 5.

**Figure 2 ijms-21-00326-f002:**
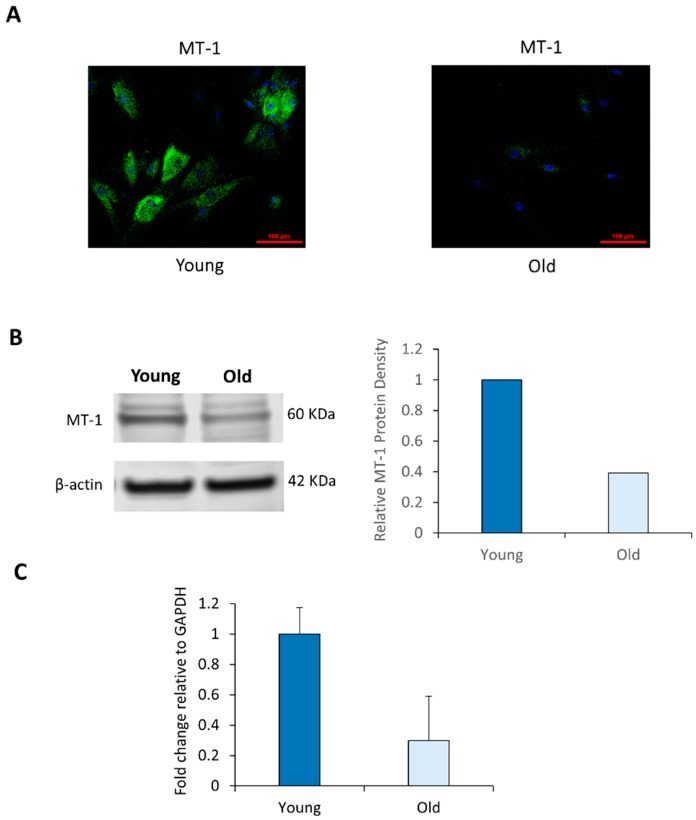
Expression of melatonin MT-1 receptor in adult NHDF declines with age. (**A**) MT-1 localization was visualized using specific antibodies as described in Methods. (**B**) Protein analysis by Western blot and (**C**) mRNA analysis by RT-PCR of fibroblast extracts derived from young and old subjects. Error bars are SEM.

**Figure 3 ijms-21-00326-f003:**
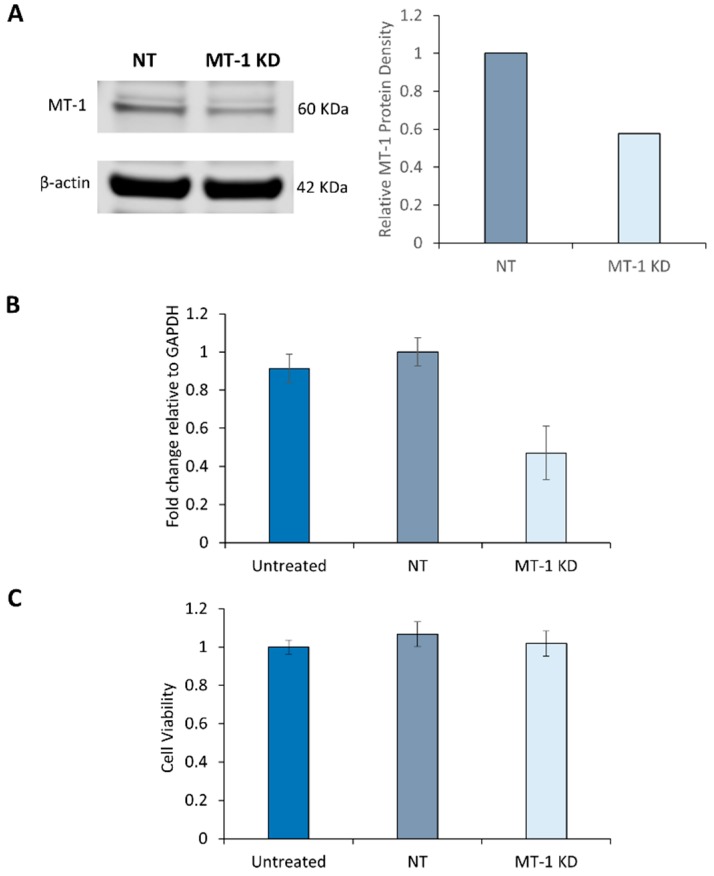
Knockdown of MT-1 receptor. (**A**) Young NHDF were electroporated with MT-1 receptor specific siRNA. Protein levels were measured by Western blot 72 h post-electroporation and (**B**) mRNA levels by RT-PCR 48 h post-electroporation. NHDF were electroporated with non-targeting (NT) siRNA for comparison. (**C**) The viability of electroporated NHDF was compared with the untreated cells and assessed by Alamar Blue staining. Error bars are SEM. *N* = 4.

**Figure 4 ijms-21-00326-f004:**
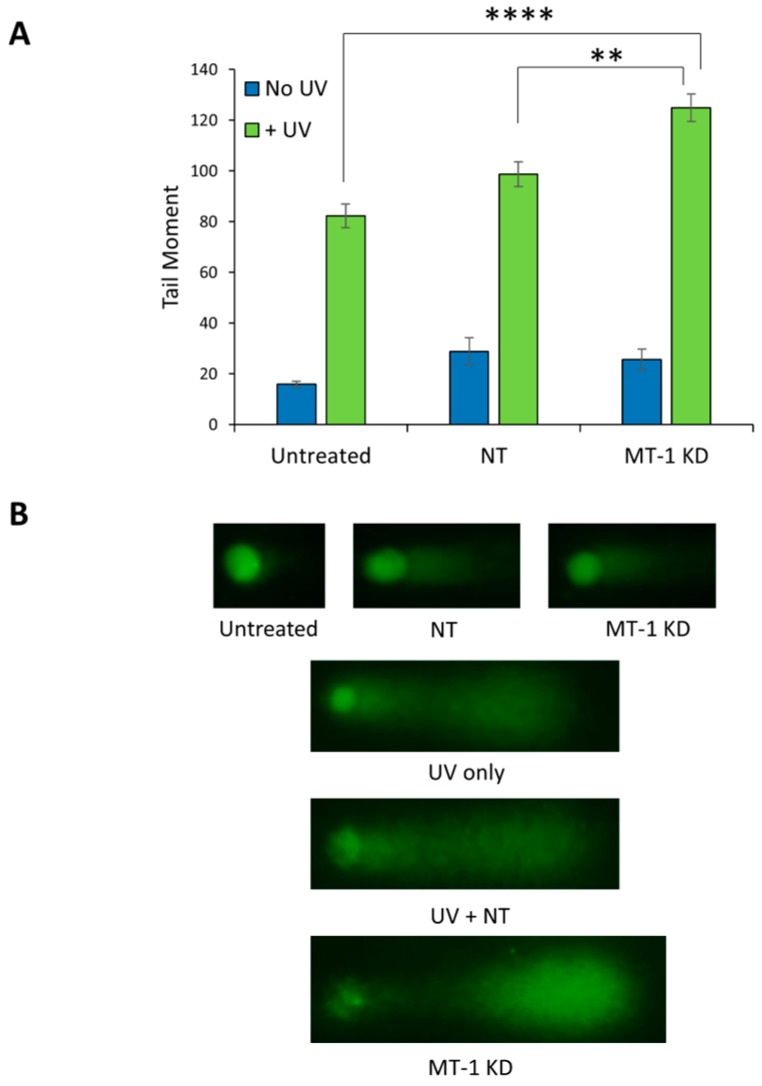
DNA damage is elevated in MT-1 knockdown NHDF exposed to UVA (5 J/cm^2^) and UVB (40 mJ/cm^2^). The cells were then assayed for DNA damage by subjecting the cells embedded in agarose to electrophoresis and then staining for DNA with SYBR Gold. An increased length of DNA staining in the immunocytochemistry figures (“comet”) indicates increased damage. Panel (**A**) shows the graphical representation of the “comet tail” and, panel (**B**) shows the actual immunocytochemistry photos (*** *p* < 0.0001, ** *p* < 0.01). Error bars are SEM. *N* = 18.

**Figure 5 ijms-21-00326-f005:**
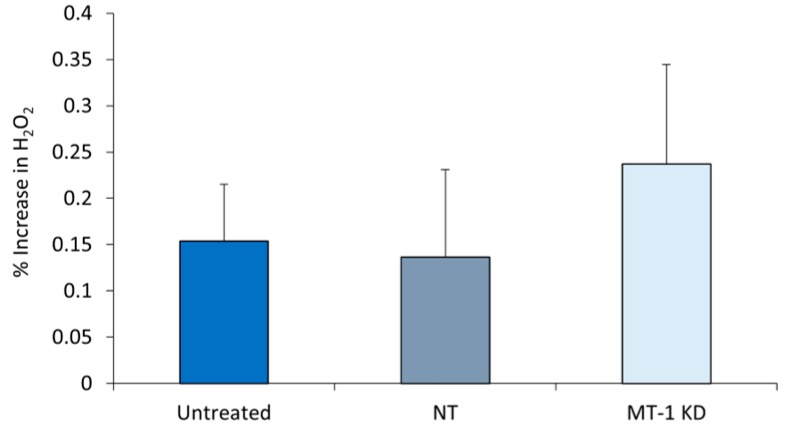
H_2_O_2_ generation is elevated in MT-1 knockdown NHDF exposed to UV. NHDF were treated with non-targeted siRNA (NT), targeted siRNA (MT-1 KD) or untreated, and exposed to UV (5 J/cm^2^ of UVA + 50 mJ/cm^2^ of UVB). The amount of H_2_O_2_ generated were quantitated as described in Methods and normalized to the total cellular protein. Error bars are SEM. *N* = 4.

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
