# Peer review of "Age Associated Decrease of MT-1 Melatonin Receptor in Human Dermal Skin Fibroblasts Impairs Protection Against UV-Induced DNA Damage"

_ijms, 2020, doi:10.3390/ijms21010326_

Round 1

Reviewer 1 Report

The authors demonstrated a critical role of MT-1 receptor in protecting skin cells from UV-induced DNA damage that might suggest that the melatonin pathway plays a pivotal role in skin aging and damage. They also proposed a possibility to develop a new approach for decelerating skin aging by modulating the expression of melatonin receptors in human skin.

I recommend the publication of this manuscript in its present form.

Author Response

Thanks you for reviewing our paper.

Reviewer 2 Report

In their study entitled “Age associated decrease of MT-1 melatonin receptor in human dermal skin fibroblasts impairs protection against UV-induced DNA damage” the authors showed that in normal human dermal fibroblasts and normal human epidermal keratinocytes melatonin is capable of stimulating PER1 expression in dose-dependent manner. In addition, they confirmed the expression of MT-1 receptor in human skin fibroblasts and revealed age-dependent decreasing in MT-1 level in mature fibroblasts as well as the important role of MT-1 receptor in protecting the skin cells from UV-induced DNA damage.

This is interesting topic however, I believe the manuscript would profit if in the Introduction authors write briefly about arylalkylamine N-acetyltransferase (AANAT) enzyme as critical regulatory element of melatonin synthesis.

Some additional minor points:

In Figure 2 B and Figure 3 A, the authors may provide also a graphical representation of the western blot results. This will help to better view the protein expression results.

Page 8, line 220 “the melatonin to the cytoplasm [4545]” there is twice reference 45, please correct.

Page 8, line 221 “NRF2 transcription in the nucleus, possibly inhibit the proteasome [4646]” there is twice deference 46, please correct.

Page 13, line 447, reference 35. K. KleszczyÅ„ski, B. Bilska, A. Stegemann, D.J. Flis, W. Ziolkowski, E. Pyza, T.A. Luger, R.J. Reiter, M. Böhm, A.T. Slominski, Int. J. Mol. Sci., 2018, 19, E3786. Please provide the title of the article.

Author Response

We added information about AANAT in the introduction Graphical representations of the western blots have been added Reference 45 and 46 listings in text were corrected Title was added to reference 35